# The Rationale and Design of the KOSovan Acute Coronary Syndrome (KOS-ACS) Registry

**DOI:** 10.3390/diagnostics14141486

**Published:** 2024-07-11

**Authors:** Gani Bajraktari, Shpend Elezi, Ibadete Bytyci, Pranvera Ibrahimi, Genc Abdyli, Edita Pllana-Pruthi, Rona Karahoda, Arlind Batalli, Afrim Poniku, Mentor Shatri, Drilon Gashi, Artan Bajraktari, Faik Shatri, Michael Y. Henein

**Affiliations:** 1Clinic of Cardiology, University Clinical Centre of Kosova, 10000 Prishtina, Kosovo; selezi@hotmail.com (S.E.); i.bytyci@hotmail.com (I.B.); pranvera_i86@hotmail.com (P.I.); gencabdyli@yahoo.com (G.A.); editapllana@hotmail.com (E.P.-P.); arlindbatalli@hotmail.com (A.B.); afrimponiku@hotmail.com (A.P.); mentorshatri9@gmail.com (M.S.); gashidriloni@hotmail.com (D.G.); artanbajraktari@hotmail.com (A.B.); faikshatri@hotmail.com (F.S.); 2Medical Faculty, University of Prishtina, 10000 Prishtina, Kosovo; 3Department of Public Health and Clinical Medicine, Umeå University, 901 87 Umeå, Sweden; michael.henein@umu.se; 4Research Unit, Heimerer College, 10000 Prishtina, Kosovo; rona.karahoda@kolegji-heimerer.eu

**Keywords:** acute coronary syndrome, registry, Kosovo, in-hospital mortality, outcomes

## Abstract

The KOSovan Acute Coronary Syndrome (KOS-ACS) Registry is established as a prospective, continuous, nationwide, web-based registry that is operated online. The KOS-ACS registry is designed with the following objectives: (1) to obtain data on the demographic, clinical, and laboratory characteristics of ACS patients treated in Kosovo; (2) to create a national database with information on health care in ACS patients treated in Kosovo; (3) to identify the national features of associations between ACS characteristics and clinical outcomes, including mortality, complications, the length of hospital stay, and the quality of clinical care; and (4) to propose a practical guide for improving the quality and efficiency of ACS treatment in Kosovo. The Kosovo Society of Cardiology and University of Prishtina will be responsible for the development of the KOS-ACS registry and centralized data analysis at the national level. The KOS-ACS Registry will enroll all patients admitted, at any of the registered clinical centers, with the diagnosis of ACS and who will be clinically managed at any of the Kosovo hospitals. Data on patient demographics, clinical characteristics, previous and hospital drug treatment, and reperfusion therapy will be collected. The type of ACS (unstable angina, NSTEMI, or STEMI) will also be clearly defined. The time from first medical contact to balloon inflation (FMC-to-balloon) and door-to-ballon time will be registered. In-hospital death and complications will be registered. Data on the post-hospital primary outcome (MACE: cardiac death, all-cause mortality, hospitalization, stroke, need for coronary revascularization) of patients, at 30 days and 1 year, will be included in the registry.

## 1. Introduction

Acute coronary syndrome (ACS), which includes unstable angina, non-ST segment elevation myocardial infarction (NSTEMI), and ST segment elevation myocardial infarction (STEMI), is a common manifestation of atherosclerotic cardiovascular disease (CVD), and its recognition has a direct impact on patients’ morbidity and mortality [1,2,3,4,5]. Numerous national registries, worldwide, do exist, and many include patients with ACS [6,7,8,9,10,11,12,13,14,15]. The main goal of these registries is to fill the gap between real-world data, on the treatment of patients with ACS, and current relevant clinical guidelines, derived from randomized controlled trials [16,17]. Other important objectives of ACS registries are to investigate the clinical characteristics of patients with ACS in different countries/regions. Thus, the most important rationale of ACS registries is evaluating and optimizing national ACS practice strategies in light of the available up-to-date strong international evidence. This rationale has been useful in many countries [18,19,20,21,22,23,24,25,26], having identified wide geographic variations in the management of patients with STEMI, especially in receiving timely primary PCI [27]. 

Kosovo is the youngest European country and does not have any prospective designed registry for ACS. Some observational single-center data in different time periods have shown that the treatment of patients with ACS in Kosovo [28,29,30] did not strictly comply with current clinical guidelines, resulting in significantly sub-optimum clinical outcomes. Therefore, the Board of Kosovo Society of Cardiology, in its role to rectify such a situation, has approved establishing a National ACS Registry. The results of such a registry are expected to have an important impact on health care providers of the country, persuading them to take adequate measures for improving the clinical care of patients with ACS. This National Registry is described as the KOSovan Acute Coronary Syndrome (KOS-ACS) Registry. 

The aim of this article is to discuss the objectives and design of the KOS-ACS Registry which should also be of interest to other nations. 

## 2. Description of KOS-ACS Registry 

### 2.1. Objectives

The KOS-ACS registry is designed with the following objectives: (1) to obtain data on the demographic, clinical, and laboratory characteristics of ACS patients treated in Kosovo; (2) to create a national database with information on health care in ACS patients treated in Kosovo; (3) to identify the national features of associations between ACS characteristics and clinical outcomes, including mortality, complications, the length of hospital stay, and the quality of clinical care; and (4) to propose a practical guide for improving the quality and efficiency of ACS treatment in Kosovo (Figure 1). 

### 2.2. Desing of the KOS-ACS Registry 

The KOS-ACS Registry is established as a prospective, continuous, nationwide, web-based registry that is operated online (www.kosacs.com, access on 1 June 2024). The design of the KOS-ACS Registry is based on the current clinical guidelines’ points on the diagnosis and treatment of ACS [31]. 

Only members of the registry, who will have username, a unique identification number, and password, will have access to the registry. The menus of the webpage are designed to minimize the number of keyboard errors. Several technical measures are introduced to maximize the accuracy of data. The web interface of the KOS-ACS contains these forms with the following titles: (1) personal data of ACS patients; (2) history of present event of ACS; (3) past relevant history; (4) risk factors for CVD; (5) results of physical examination; (6) results of echocardiography; (7) results of laboratory tests; (8) details of intervention, if happened; (9) prior therapy; (10) details of medical treatment of ACS; (11) recommendations at discharge; and (12) complications and clinical outcomes. 

### 2.3. The Developers of the KOS-ACS Registry 

The Kosovo Society of Cardiology and University of Prishtina will be responsible for the development of the KOS-ACS Registry and centralized data analysis at the national level. The KOS-ACS was established in November 2023 by researchers and cardiologists, members of the Kosovo Society of Cardiology, with the support of information technology specialists. It will start collecting data for ACS on 1 June 2024. 

### 2.4. Participation 

Participation in the KOS-ACS is voluntary and free-of-charge. Any clinical center that provides health care for ACS patients, both in the private and public health care system in Kosovo, is entitled to participate in the KOS-ACS Registry by sending a request to the technical support team of the KOS-ACS. In December 2023, all centers were invited to take part in the KOS-ACS by the Kosovo Society of Cardiology, and all agreed to participate. The registry will have one principal investigator and ten coordinators (one per each center).

### 2.5. Patients 

The KOS-ACS Registry will enroll all patients admitted, at any of the registered clinical centers, with the diagnosis of ACS according to conventional European Guidelines [31] and who will be clinically managed at any of the Kosovo hospitals including the University Clinical Centre of Kosova in Prishtina, six Regional Hospitals in six Kosovo districts, and three private hospitals (Appendix A). The enrollment of patients is planned to start in May 2024 and to continue for at least two years, with potential extension (Figure 2). Inclusion criteria comprise the following: (1) any type of ACS (unstable angina, NSTEMI, or STEMI) as a presumptive diagnosis and (2) patients age ≥ 18 years, irrespective of gender. The exclusion criteria will be (1) symptoms considered as ACS at admission which were not consistent with acute cardiac ischemia and (2) patients with ACS accompanied by a significant comorbidity, such as any trauma, traffic accident, or severe gastrointestinal bleed/operation or procedure directly before admission (Figure 1). 

### 2.6. Data Elements

The key data elements and definitions of the KOS-ACS Registry database were developed using the 2023 ESC Guidelines for the management of ACS [31] and the ACVC-EAPCI EORP STEMI Registry of the ESC [32].

Data on patient demographics, clinical characteristics, previous and hospital drug treatment, and reperfusion therapy will be collected. All conventional and new risk factors will also be part of the questionnaire (Appendix A) patients will be requested to fill out. The type of ACS (unstable angina, NSTEMI, or STEMI) will also be clearly defined. The time from first medical contact to balloon inflation (FMC-to-balloon) and door-to-ballon time will be registered. In-hospital death and complications will be registered. Data on the post-hospital primary outcome (MACE: cardiac death, all-cause mortality, hospitalization, stroke, need for coronary revascularization) of patients, at 30 days and 1 year, will be included in the registry (Figure 3). 

### 2.7. Data Collection

All centers participating in the KOS-ACS Registry will be asked to include all patient’s inclusion/exclusion criteria for ACS based on the recent ESC Guideline for ACS [31] or future eventual guidelines delivered by ESC. The source of patient data will be hospital charts for admitted patients with ACS in all centers. A detailed user manual will be developed, to help participants in the registry, and will be available on the KOS-ACS Registry website. 

In each participating center, at least one physician will be trained to log patient data into the registry. The content of data-entry web forms will be simple and user friendly. Every 3 months, experts from the KOS-ACS Registry will check the validity of entered data by reconciling randomly selected records with the data of patients’ hospital charts. 

### 2.8. Data Security 

Database and web security issues are important for registries. The KOS-ACS website has been designed in a way that allows all users to be assigned with a unique username/password combination which is mandatory to log on to the KOS-ACS Registry. All transactions will be recorded automatically in the web server’s log. All data entered on the website will be protected by a password on a safe server of the KOS-ACS Registry. Subject identification will be possible only at the local study site, and participating centers will be able to review and modify patient data. ACS patients’ data added to the KOS-ACS Registry may be updated but cannot be removed. The entered data will be stored on the central database on the central server at KOS-ACS Registry. These measures are undertaken to ensure the confidentiality and security of the data. 

### 2.9. Ethical Aspects 

The study protocol of the KOS-ACS Registry, including patient information and consent forms has been reviewed and approved by the Ethics Commission of the Kosovo Doctors Chamber and by the Ethics Committee of the University of Prishtina. All patients must give informed consent before entering their personal and clinical data in the KOS-ACS Registry. The standard informed consent form will be available on the KOS-ACS Registry website, in order to allow the local coordinator to obtain consent. Patients will give informed consent after transfer from an intensive care unit (ICU) to a cardiac/coronary care unit. In the case of death in the ICU, consent to use the patient’s data anonymously will be given by a patient’s relative. The appropriate measures are used to guarantee maximum data confidentiality. All patient-related clinical data will be anonymized locally. 

## 3. Discussion

National medical registries have become a well-established practice in many countries and among different academic organizations. The main benefit of medical registries is the transparent and detailed documentation of a patient’s medical condition, risk factors, clinical presentation, management pathway, and clinical outcome. Such an approach allows for an accurate evaluation of the national management strategies and potential deviation from international guidelines. It also allows for risk factors’ comparison and clinical outcome between different countries in and away from the same geographical region. The first published registry on myocardial infarction was from Warsaw emergency service in 1979 [33], followed by many registries from different countries, worldwide. The most recently published is the national Moroccan registry of ST-elevation MI [34]. Kosovo is a small-sized country, recently established with steadily developing infrastructure and economy. Likewise, medical services have been developing in different specialties including cardiovascular disease. Primary coronary intervention has now nationally matured with a clinical outcome comparable to other countries in the region [32,35,36]. The optimum control and prevention of coronary artery disease and ischemic heart disease in Kosovo, however, remain a challenge since the clinical service provided is not nationally uniform but depends on individual center expertise and facilities. In addition, the coronary disease prevention strategy in Kosovo has not matured yet. The Kosovo Society of Cardiology, established in 2002, has already made a plausible scientific and academic success with many cardiologists obtaining higher degrees from international universities and returning to support the local clinical services and bring international knowledge into daily practice for their patients. Despite this, there is a lack of a formal structured documentation system in the form of a registry which should assist clinicians and scientists to study, in detail, the nature of national Kosovo coronary artery disease, in terms of the pattern of presentation, contributing risk factors, and management pathways. Also, such a registry should help in conducting an accurate comparison of the latter information with respective ones from other nations, in light of the international available guidelines’ recommendations. Once achieved, the results should assist in drawing an accurate clinical road map for managing coronary artery disease in Kosovo.

The Kosovo Society of Cardiology has agreed to establish a national registry for coronary artery disease, the summary of which is described in this report, and the results of its analysis should assist in achieving the registry objective, in particular the evaluation of coronary disease services against published clinical guidelines and comparing its details with respective ones from other nations.

## Figures and Tables

**Figure 1 diagnostics-14-01486-f001:**
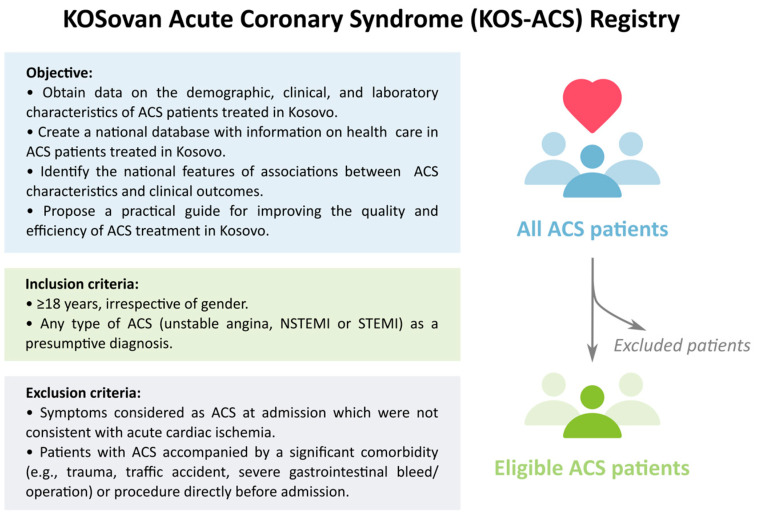
Objective, inclusion and exclusion criteria of the study patients.

**Figure 2 diagnostics-14-01486-f002:**
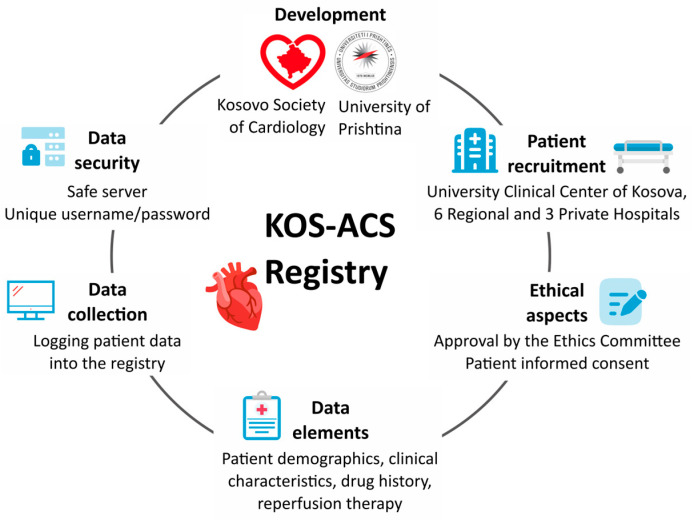
The development plan of the registry.

**Figure 3 diagnostics-14-01486-f003:**
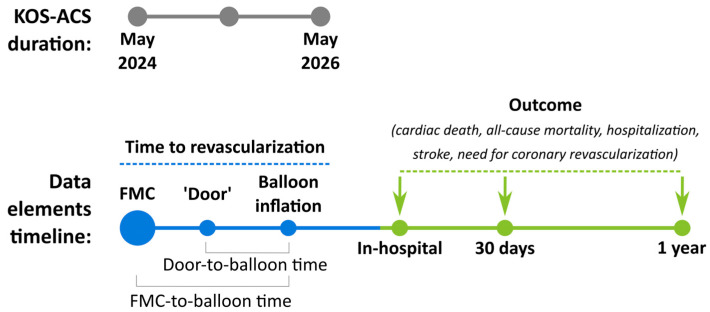
The follow-up plan of the registry.

## Data Availability

The data underlying this article will be shared on reasonable request to the corresponding author.

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
