# Peer review of "The Rationale and Design of the KOSovan Acute Coronary Syndrome (KOS-ACS) Registry"

_diagnostics, 2024, doi:10.3390/diagnostics14141486_

Round 1
Reviewer 1 Report
Comments and Suggestions for Authors
Bajraktari et al. adequately describe the purpose of their register in this communication. The article is well written and outlines how the register will be developed and edited. It certainly does not add anything new to the current literature, but provides a good perspective on how a register should be developed.
However, I have a few minor points that should be implemented:
1. figures 2 and 3 described in the article are not present, so they should be added
2. An accurate list of the data that will be collected should be provided (also as supplementary material) with the article
3. A list of participating centres should be given
Author Response
Response to Reviewer 1
Bajraktari et al. adequately describe the purpose of their register in this communication. The article is well written and outlines how the register will be developed and edited. It certainly does not add anything new to the current literature, but provides a good perspective on how a register should be developed.
However, I have a few minor points that should be implemented:
- Figures 2 and 3 described in the article are not present, so they should be added
Response:
Thank you for your suggestion. It was an error during the uploading these figures, which now we included.
- An accurate list of the data that will be collected should be provided (also as supplementary material) with the article
Response:
Thank you for your suggestion. We now included it as a supplementary material.
- A list of participating centers should be given
Response:
Thank you for your suggestion. We now included the list of centers as a supplementary material.
Reviewer 2 Report
Comments and Suggestions for Authors
In this manuscript, Bajraktari et al. describe the rationale for the design of a registry in Kosovo, the first of its kind, for acute coronary disease. This is a very noble undertaking and it will be exciting to see the development of this in the future. This reviewer had one question. Regarding the construction of the registry, have the authors thought about building a biobank with the corresponding samples? Collecting multiple sample types (e.g. whole blood, serum, plasma, tissues) from these patients along with the registry information on disease type, treatment, demographics, etc. will provide an important and significant tool for future research and clinical studies.
Author Response
Response to Reviewer 2
In this manuscript, Bajraktari et al. describe the rationale for the design of a registry in Kosovo, the first of its kind, for acute coronary disease. This is a very noble undertaking and it will be exciting to see the development of this in the future.
This reviewer had one question. Regarding the construction of the registry, have the authors thought about building a biobank with the corresponding samples? Collecting multiple sample types (e.g. whole blood, serum, plasma, tissues) from these patients along with the registry information on disease type, treatment, demographics, etc. will provide an important and significant tool for future research and clinical studies.
Response:
We will collect the blood samples for biobank, but this belongs to another joint project that is waiting for the Ethics decision.